# Assessment of Biological Activity of Low Molecular Weight 1,4-Benzoquinone Derivatives

**DOI:** 10.3390/biom15081162

**Published:** 2025-08-14

**Authors:** Marija Bartolić, Ana Matošević, Nikola Maraković, Irena Novaković, Dušan Sladić, Suzana Žunec, Dejan Opsenica, Anita Bosak

**Affiliations:** 1Institute for Medical Research and Occupational Health, Ksaverska cesta 2, 10001 Zagreb, Croatia; mbartolic@imi.hr (M.B.); amatosevic@imi.hr (A.M.); nmarakovic@imi.hr (N.M.); suzana@imi.hr (S.Ž.); 2Institute of Chemistry Technology and Metallurgy, University of Belgrade, Njegoševa 12, 11000 Belgrade, Serbia; irena.novakovic@ihtm.bg.ac.rs (I.N.); dejan.opsenica@ihtm.bg.ac.rs (D.O.); 3Faculty of Chemistry, University of Belgrade, Studentski trg 12-16, 11158 Belgrade, Serbia; dsladic@chem.bg.ac.rs; 4Centre of Excellence in Environmental Chemistry and Engineering, Njegoševa 12, 11000 Belgrade, Serbia

**Keywords:** AChE/BChE inhibition, flexible molecular docking, BBB prediction, chelation, antioxidant screening

## Abstract

In this paper, we aimed to evaluate whether simple, low molecular mass benzoquinone derivatives, featuring different substituents in *para*- and *meta*-position relative to the *tert*-butyl group, possess biological activities against major targets associated with Alzheimer’s disease. The 1,4-benzoquinone derivatives studied herein inhibited both cholinesterases in the micromolar concentration range, generally showing a preference for butyrylcholinesterase over acetylcholinesterase; formed complexes with biometal ions Fe^2+^, Cu^2+^ and Zn^2+^; and displayed a certain BACE1 inhibition. Moreover, the tested compounds displayed certain antioxidant activity via either electron transfer or hydrogen atom transfer mechanisms. The antioxidant capacity of the unsubstituted *tert*-butyl-1,4-benzoquinone (compound **1**) was three times lower than that of the standard antioxidant BHT, while 2,6-disubstituted derivatives (compounds **15** and **7**) exhibited peroxyl radical scavenging activity comparable to that of Trolox. Taken together with in silico-predicted low toxicity, good intestinal absorption and favorable oral bioavailability, the presented 1,4-benzoquinone derivatives are promising scaffolds for the design of more complex molecules with enhanced cholinesterase and BACE1 inhibitory activity. Furthermore, they could serve as functional substituents in other structural scaffolds to combine and enhance their biological activities.

## 1. Introduction

Alzheimer’s disease (AD) is a chronic neurodegenerative disease and the leading contributor to global dementia cases. It is characterized by the progressive deterioration of memory, communication capacity, spatial orientation, self-care capacity, and patient’s cognitive functions [1]. The cholinergic hypothesis was the first to provide insight into the pathophysiological mechanisms underlying cognitive decline in AD, focusing on the degeneration of cholinergic neurons and the depletion of acetylcholine (ACh) levels in the brain [2,3,4]. Consequently, the inhibition of acetylcholinesterase (AChE), the main enzyme responsible for ACh hydrolysis, emerged as a central strategy in AD drug development, aiming to elevate ACh levels in the brain [5,6]. Over time, the cholinergic hypothesis was expanded to include butyrylcholinesterase (BChE), based on findings that AChE activity decreases in the cortex of AD patients as the disease progresses, while BChE activity remains constant or increases up to 120%, assuming a more prominent role in ACh hydrolysis [7]. Another major theory elucidating AD pathogenesis is the amyloid hypothesis, which focuses on the release and self-aggregation of β-amyloid (Aβ) peptides via sequential cleavage of amyloid precursor protein (APP) by β-secretase (BACE1) and γ-secretase [8,9]. However, AD is a complex, multifactorial condition whose pathogenesis remains only partially understood. In addition to the previously described pathophysiological mechanisms—most notably those contributing to dementia as the hallmark clinical manifestation—a variety of additional pathological processes have been identified, giving rise to new hypotheses regarding the origins and progression of the disease. One such theory is the tau hypothesis, which proposes that the abnormal hyperphosphorylation of the tau protein leads to its dissociation from microtubules, resulting in microtubule destabilization, subsequent degradation, and the formation of neurofibrillary tangles (NFTs). The accumulation of NFTs disrupts the intracellular transport of nutrients and essential molecules, ultimately impairing synaptic function and contributing to neuronal dysfunction and cognitive decline [10,11]. According to the oxidative stress hypothesis, the overproduction of reactive oxygen species (ROS) is a major contributing factor to AD pathogenesis, due to the brain’s increased vulnerability to oxidative damage. ROS can impair neuronal function by damaging essential cellular biomolecules such as lipids, proteins, and nucleic acids [12]. Numerous studies strongly suggest that neuroinflammation, characterized by the activation of glial cells, particularly microglia and astrocytes, plays a significant role in the progression of AD. Namely, in response to the presence of Aβ and tau aggregates in the brain, microglia become activated, migrate to amyloid plaques, and attempt to clear Aβ via phagocytose. However, the immune response, if prolonged, may become detrimental, contributing to synaptic dysfunction and neuronal damage [13]. The complex network of AD hallmarks also includes the dyshomeostasis of biometal ions Fe^2+^, Cu^2+^ and Zn^2+^, due to their involvement in the generation of ROS, association with neuroinflammation, ability to enhance the production of Aβ and promote their aggregation by direct binding to Aβ and tau proteins [14,15].

To date, the United States Food and Drug Administration (FDA) has approved eight medications for the treatment of AD, which can be categorized into two groups: symptomatic therapies and disease-modifying treatments. Symptomatic treatments include the cholinesterase inhibitors donepezil, galantamine, rivastigmine and benzgalantamine (a prodrug of galantamine), all of which are approved for the treatment of mild-to-moderate AD. Additionally, memantine, an NMDA receptor antagonist that modulates glutamatergic activity, and the donepezil-memantine combination are approved as therapies for moderate-to-severe stages of the disease [16]. In 2023 and 2024, lecanemab and donanemab were approved as the first disease-modifying agents for AD treatment. These monoclonal antibodies target aggregated β-amyloid, aiming to reduce amyloid plaque burden and slow cognitive decline [17,18]. However, considering the multifactorial nature of AD, the main limitation of all currently approved therapies is that they target only one hallmark of the disease, and therefore such single-target approaches are unlikely to fully address the complexity and heterogeneity of its pathophysiology. To overcome this, an innovative strategy has emerged involving the development of multi-target-directed ligands (MTDLs)—small molecules designed to incorporate two or more pharmacophores capable of simultaneously interacting with multiple targets involved in AD pathogenesis [19,20]. Numerous MTDL candidates have been synthesized and evaluated, with some advancing to clinical trials [20]. This approach requires the precise optimization of the pharmacodynamic balance between different active moieties and strict control of molecular weight (ideally <400–600 Da), which is a key determinant for effective penetration across the blood–brain barrier (BBB) and central nervous system activity [21,22].

1,4-Benzoquinones, commonly known as para-quinones, are the simplest naturally occurring quinones found in various biological systems including plants, animals and bacteria. They are recognized for their antioxidant potential, and their structural motif is present in the biochemical cofactor coenzyme Q10, which plays a significant role in the mitochondrial oxidative phosphorylation pathway [23]. In medicine, drugs containing 1,4-benzoquinone scaffolds have shown therapeutic relevance in several diseases. For instance, idebenone, a synthetic derivative of coenzyme Q10, is used to treat visual impairment in Leber’s hereditary optic neuropathy. Other compounds—such as antroquinonol, mitoquinone and vatiquinone—are under investigation for the treatment of hyperlipidemia, non-small cell lung cancer, hepatitis C, Parkinson’s disease, Rett syndrome, and methylmalonic aciduria and the prevention of retinopathy [16]. Studies have demonstrated that incorporating the 1,4-benzoquinone core in various molecular scaffolds can yield compounds with potent biological activity relevant to AD. These include strong inhibitory effects on AChE, BChE and BACE1; the ability to prevent or reverse Aβ aggregation; and the biometal chelation and regulation of biometal-free and biometal-bound Aβ aggregation, showing promising efficacy in both in vitro and in vivo studies [23,24,25,26,27,28,29]. Among them, memoquin stands out as a prominent multitarget compound, exhibiting the nanomolar inhibition of AChE and micromolar activity against BACE1, Aβ self-aggregation, and AChE-induced Aβ aggregation. Considering that 1,4-benzoquinones represent well-established pharmacophores with board therapeutic potential, their further chemical derivatization may yield promising candidates for AD treatment [23].

In this study, we aimed to evaluate whether simple low molecular mass 1,4-benzoquinone derivatives exhibit biological activity against key targets implicated in AD pathogenesis. We selected a series of fifteen 1,4-benzoquinone derivatives, each bearing substituents containing sulfur or nitrogen atoms. These heteroatoms were chosen as we hypothesized that they could significantly influence the electron density of the quinone core, thereby affecting its interaction with amino acid residues within enzyme active sites. Furthermore, electron-donating substituents are expected to enrich the quinone ring’s electron density, rendering the system less susceptible to nucleophilic attack by biological nucleophiles—an effect that may contribute to reduced toxicity [30,31].

The selected series, comprising nine novel compounds and six previously reported analogs (Figure 1), was evaluated in vitro against several major pathological features of AD, including acetylcholine depletion, amyloid-β aggregation, biometal dyshomeostasis, and oxidative stress. Compounds were tested for inhibitory activity against human AChE, BChE and BACE1. Metal-chelating properties were assessed using Fe^2+^, Cu^2+^ and Zn^2+^, and antioxidant capacity was evaluated using the FRAP and ORAC assays. In addition to experimental assays, in silico methods were employed to predict the binding modes of the compounds to AChE and BChE, as well as to estimate their toxicity and oral bioavailability, based on calculated physicochemical properties.

## 2. Materials and Methods

### 2.1. Materials

All reagents and solvents used for the synthesis were purchased from commercial sources (Fluka, Sigma, Aldrich or Merck from Merck KGaA, Darmstadt, Germany or Acros Organics from Thermo Fisher Scientific Inc., Waltham, MA, USA). All solvents were distilled before use. Reaction progress was monitored by thin-layer chromatography (TLC) using Supelco TLC aluminum sheets precoated with Silica gel 60 and a UV indicator (254 nm). Preparative TLC was performed on Supelco silica gel 60 GF254 (Merck KGaA, Darmstadt, Germany) with a UV-active indicator and appropriate mobile phase, as described in the corresponding synthetic procedures (details available in the Appendix A). NMR spectra were recorded in deuterochloroform (CDCl_3_) on a Bruker Avance III (500 MHz instrument for ^1^H NMR and 125 MHz for ^13^C NMR) (Billerica, MA, USA). Chemical shifts were reported in parts per million (ppm) using tetramethylsilane as the internal standard, the coupling constants (*J*) were given in Hz, and the multiplets were designated as follows: singlet (s), broad singlet (bs), doublet (d), double doublet (dd), triplet (t), and multiplet (m). The ESI-MS spectra of the synthesized compounds were recorded on an Agilent Technologies 1200 Series (Santa Clara, CA, USA) instrument equipped with a Zorbax Eclipse Plus C18 (Santa Clara, CA, USA) (100 mm × 2.1 mm i.d., 1.8 μm) column and a DAD detector (190–450 nm) in combination with an Agilent Technologies 6210 Time-Of-Flight LC-MS (Santa Clara, CA, USA) instrument in positive ion mode with CH_3_CN/H_2_O 1/1 with 0.2% HCOOH as the carrying solvent solution. Samples were dissolved in MeOH (HPLC grade purity). The capillary voltage = 4 kV, gas temperature = 350 °C, drying gas flow rate = 12 L min^−1^, nebulizer pressure = 45 psi, and fragmentor voltage = 70 V were used.

All measurements of cholinesterase activity were performed in 0.1 M sodium phosphate buffer (pH 7.4). Acetylthiocholine (ATCh), used as a substrate, and thiol reagent 5,5′-dithiobis (2-nitrobenzoic acid) (DTNB) were purchased from Sigma–Aldrich, St. Louis, MO, USA. ATCh was dissolved in water and DTNB in sodium phosphate buffer. Stock solutions of compounds were prepared in DMSO (Gram-mol, Zagreb, Croatia), and all further dilutions were carried out in water. As the source of AChE and BChE, purified native human BChE and recombinant human AChE, both kindly provided by Dr Florian Nachon (Département de Toxicologie, Armed Forces Biomedical Research Institute (IRBA), Brétigny-sur-Orge, France), were used. Both enzymes were diluted in a phosphate buffer containing 0.1% BSA, while further dilutions were in a phosphate buffer with 0.01% BSA.

For the determination of BACE1 activity, a human recombinant β-secretase (BACE1) (cat. no. S4195, Sigma Aldrich, St. Louis, MO, USA) as the source of the enzyme and 7-Methoxycoumarin-4-acetyl-[Asn670, Leu671]-Amyloid β/A4 Precursor Protein 770 Fragment 667–676(2,4-dinitrophenyl)Lys-Arg-Arg amide (cat. no. A1472, Sigma Aldrich, St. Louis, MO, USA) as the substrate were used. All measurements were conducted in 50 mM sodium acetate buffer (pH 4.5) with 0.005% Triton X-100. Commercially available β-secretase Inhibitor III (cat. no. 565780, EMD Millipore Corp., Billerica, MA, USA) was used as positive control.

For metal chelation studies, metal salts ZnCl_2_, CuCl_2_ × 2H_2_O, and FeCl_2_ × 4H_2_O were used (Sigma Aldrich, St. Louis, MO, USA).

All chemicals for FRAP antioxidant activity measurements were purchased from Sigma Aldrich (St. Louis, MO, USA), except for FeCl_3_ and tripyridyltriazine (TPTZ), which were purchased from Kemika, Zagreb, Croatia, and Fluka, Buchs, Switzerland, respectively.

For the determination of antioxidant capacity by ORAC method, the commercially available OxiSelect™ Oxygen Radical Antioxidant Capacity (ORAC) Activity Assay kit was used (cat. no. STA-345, Cell Biolabs, Inc., San Diego, CA, USA).

### 2.2. Synthesis

The new compounds were obtained in a one-pot, two-step reaction, starting from *tert*-butyl-1,4-benzoquinone and the corresponding thiol in an argon atmosphere, to prevent the oxidation of thiol reagents. The reaction started with Michael’s addition, followed by the oxidation of the intermediates by the excess of TBQ. After the completion of the reaction, standard work-up procedures were performed, and the products were separated and purified by preparative TLC. All compounds were characterized by nuclear magnetic resonance (NMR) and electrospray ionization mass spectrometry (ESI-MS). Detailed reaction schemes, spectral data, and copies of NMR spectra are given in the Appendix A. The synthesis and characterization of six compounds (**1**, **11**–**15**) used to complete the series were published previously [30,31,32,33]. Since the studied 1,4-benzoquinones are also conjugated carbonyl compounds that readily undergo nucleophilic attack, especially in the presence of acid, which is usually present in reversed phase conditions, the purities of the examined compounds are based on NMR spectra and HRMS data, along with accompanying chromatograms obtained during analysis. For all compounds, the purity is greater than 95%.

### 2.3. Inhibition of Cholinesterases

Enzyme activity measurements were carried out spectrophotometrically using a slightly modified Ellman method [34], as previously described [35]. The activities of AChE and BChE were measured at five different ATCh concentrations (0.05–0.50 mM) in the absence (*v*_0_) and presence (*v*_i_) of different benzoquinone concentrations (*i*); final inhibitor concentrations were in range 10–400 µM, depending on the compound potency required to achieve 20–80% inhibition of enzyme activity. At least three inhibitor concentrations were used at each substrate concentration in at least three experiments. To calculate the enzyme inhibitor dissociation constants (*K*_i_), the Hunter–Downs equation and the linear regression analysis were used [35]:*K*_i,app_ = (*v*_i_∙*i*)/(*v*_0_ − *v*_i_) = *K*_i_ + *K*_i_/*K*_S_ *s*
where *K*_i,app_ stands for the apparent inhibition constant and *K*_S_ for the enzyme–substrate dissociation constant.

The final content of DMSO in measurements was up to 0.4%, and inhibitory reactions containing more than 0.1% of DMSO were corrected for control reactions containing a corresponding concentration of DMSO. No side interactions of the tested compounds with ATCh or DTNB were detected. Measurements were performed at 25 °C and wavelength λ = 412 nm on a Tecan Infinite M200Pro (Tecan Austria GmbH, Salzburg, Austria), and for all calculations, the statistical package GraphPadPrism 8 (Graph Pad Inc., San Diego, CA, USA) was used.

The selectivity index (SI) was determined using the following ratio: SI = *K*_i(AChE)_/*K*_i(BChE)_.

### 2.4. Inhbition of BACE1

BACE1 activity was determined using a slightly modified spectrofluorimetric method, as described in our previous work [36]. Stock solutions of tested compounds were prepared in DMSO, and further dilutions were carried out in 50 mM sodium acetate buffer (pH 4.5) containing 0.005% Triton X-100. The inhibition of BACE1 activity by benzoquinones was determined as a percentage from the ratio of fluorescence signals obtained from the inhibition reaction (sodium acetate buffer, BACE1, substrate, and 50 µM tested compound) and control reaction (sodium acetate buffer, BACE1, and substrate only). To eliminate any potential influence of benzoquinone and/or buffer on the intensity of fluorescence signal, non-enzymatic (buffer, substrate, and benzoquinones) and blank reactions (buffer and substrate only) were also measured and subtracted from inhibition and control reaction, respectively. As a positive control of BACE1 inhibition, commercially available β-secretase Inhibitor III (EMD Millipore Corp., Billerica, MA, USA) was used (5 µM), which inhibited over 90% of BACE1 activity, corresponding to the manufacturer’s data [37,38]. All fluorescence signal readouts were recorded at room temperature (25 °C) at zero and two hours after incubation at 37 °C, on a plate reader (SpectraMax iD3, Molecular Devices, LLC, San Jose, CA, USA) with excitation at 320 nm and emission at 405 nm.

### 2.5. Metal Chelation Studies

To determine the metal chelating ability of benzoquinones toward iron, copper, and zinc ions, metal salts ZnCl_2_, CuCl_2_ × 2H_2_O, and FeCl_2_ × 4H_2_O were used as metal sources, following a previously reported protocol [39,40]. The UV–Vis absorption spectra (wavelength range of 200–600 nm) of benzoquinones (15 μM), metal salts (30 μM), and the mixture of benzoquinones (15 μM) and metal salts (30 μM) were recorded at three incubation-time points (1, 60, and 120 min). Evidence of complex formation was determined by observing changes in the absorption spectra of the benzoquinone–metal mixtures relative to the spectra of the individual components (benzoquinone or metal salt alone). Additional confirmation of complex formation was obtained by determining differential UV/Vis spectra, as described previously [39,40].

All spectra recordings were performed using a UV–Vis spectrophotometer (Cary 300 spectrophotometer Varian, Inc., Belrose, Australia) and calculations using GraphPadPrism 8 (Graph Pad Inc., San Diego, CA, USA).

### 2.6. Antioxidant Activity

#### 2.6.1. FRAP Method

A ferric-reducing antioxidant power (FRAP) assay was used to determine the antioxidant capacity of tested benzoquinones, following a slightly modified protocol by Benzie and Strain [41,42]. The water-soluble derivative of vitamin E (Trolox) and butylated hydroxytoluene (BHT) were used as standard antioxidants. The reducing capacity of each compound was evaluated at 50 and 100 μM concentrations, selected based on the determined *K*_i_ values for AChE and BChE inhibition. All absorbance readouts were recorded at 593 nm against a blank and corrected for the value of absorbance of corresponding benzoquinone alone. All measurements were performed in three independent experiments. FRAP values were calculated using a standard curve for FeSO_4_ × 7H_2_O. All readouts were recorded on a plate reader (SpectraMax iD3, Molecular Devices, LLC, San Jose, CA, USA).

#### 2.6.2. ORAC Assay

The ability of benzoquinones to prevent excessive oxidation caused by ROS was tested using the commercially available OxiSelect™ Oxygen Radical Antioxidant Capacity (ORAC) Activity Assay kit (cat. no. STA-345, Cell Biolabs, Inc., San Diego, CA, USA), following the manufacturer’s protocol with slight adjustments concerning the antioxidant standard final concentrations. Stock solutions of benzoquinones were prepared in DMSO while 5 mM Trolox (used as an antioxidant standard) was provided by the manufacturer as a solution, and their further dilutions were carried out in 50% *v*/*v* acetone. Benzoquinones (50 µM) and Trolox (2.5–50 µM) were incubated with 1x Fluorescein Solution in 96-well black microplates at 37 °C for 30 min. After incubation time, Free Radical Initiator Solution was added. The sample reading was performed at 37 °C with an excitation wavelength of 480 nm and an emission wavelength of 520 nm. The antioxidant capacity of the compound was determined from the area under the fluorescence decay curve (AUC), from which the AUC of blank (50% acetone + 1x Fluorescein Solution + Free Radical Initiator Solution) was subtracted to obtain Net AUC. Net AUC values were compared to a Trolox standard curve and antioxidant capacity was presented as Trolox equivalent (TE) concentrations. All readouts were recorded on a plate reader (SpectraMax iD3, Molecular Devices, LLC, San Jose, CA, USA). Calculations were performed using the statistical package GraphPadPrism 8 (Graph Pad Inc, San Diego, CA, USA).

### 2.7. In Silico Studies

#### 2.7.1. Docking Studies

To elucidate the non-bonding interactions between selected ligands and human AChE and BChE, the flexible molecular docking was performed using the Biovia Discovery Studio Client v21. (Dassault Systèmes, Vélizy-Villacoublay, France). The docking procedure was based on previously described protocols [36], utilizing the crystal structures of human AChE (PDB ID: 4EY4) and BChE (PDB ID: 1P0I) [43,44]. According to the applied protocols, prior to molecular docking, ligands were constructed, minimized, and prepared, taking into consideration the possible different protonation states, isomers, and tautomers at pH 7.4. The representative pose of each of the docked ligands was chosen based on the highest consensus score predicted by the scoring functions estimating binding affinities implemented in the Biovia Discovery Studio Client v21, Score Ligand Poses protocol.

#### 2.7.2. Human Intestinal Absorption

Human intestinal absorption (HIA) was predicted using the model available on the pkCSM online platform [45], which is based on the lipophilicity of chemicals evaluated through the partition coefficient (log*P*) and their polarity determined by the calculated topological polar surface area (TPSA). This model has been demonstrated to effectively distinguish between well-absorbed and poorly absorbed compounds based on these physicochemical properties [46].

#### 2.7.3. Oral Bioavailability

To evaluate the potential of the tested 1,4-benzoquinone derivatives as orally active drugs in humans, key physicochemical properties influencing oral bioavailability were calculated using the Chemicalize 2018 platform [47]: molecular weight (MW), partition coefficient log*P*, number of hydrogen bond donors (HBD), number of hydrogen bond acceptors (HBA), number of rotatable bonds (RB) and polar surface area (PSA). The results were compared against established guidelines and recommended values of physicochemical properties for orally active drugs [48,49].

#### 2.7.4. Toxicity Profile

The toxicity of benzoquinones was estimated using in silico models incorporated into the pkCSM online platform: Ames toxicity, hepatotoxicity, and Minnow toxicity [46]. The Ames toxicity model is based on a bacterial assay that evaluates a compound’s potential to cause reverse mutations in specific strains of *Salmonella typhimurium* or *Escherichia coli*. These strains carry pre-existing mutations that prevent them from synthesizing essential amino acids, such as histidine or tryptophan. A reverse mutation restores the original gene function, allowing the bacteria to grow on selective media lacking these amino acids. The occurrence of such mutations indicates the potential mutagenic activity of the tested compound. The Ames model, built from data on over 8000 compounds, predicts whether a compound is likely to be Ames positive, i.e., mutagenic. The hepatotoxicity model predicts liver toxicity based on observed liver-related side effects from 531 compounds. It estimates whether a compound is likely to disrupt normal liver function. The Minnow toxicity predictive model, built from the data on the lethal concentration (LC_50_) values of 554 compounds, predicts the concentration of a substance required to cause the death of 50% of fathead minnows. Compounds with LC_50_ values below 0.5 mM are regarded as toxic [46].

## 3. Results and Discussion

### 3.1. Cholinesterase Inhibition

Of the fifteen compounds, fourteen were tested as inhibitors of human AChE and BChE (Table 1). Compound **11** was not tested because it precipitated upon addition to the phosphate buffer used in the assay. The inhibition potency of the compounds was tested in the range of 10–400 μM, depending on the compound. Compound **13** did not show any inhibitory activity at 400 μM concentration (the maximum inhibitor concentration used) and 0.1 mM ATCh for both AChE and BChE, respectively.

In general, all compounds reversibly inhibited both cholinesterases, and their inhibition potency was expressed as the dissociation constant, *K*_i_, of the enzyme–inhibitor complex.

The tested compounds inhibited AChE with *K*_i_ constants ranging from 21 to 136 μM. Although these values span approximately one order of magnitude, the *K*_i_ constants for most compounds were similar to that of compound **1**, which represents the structural scaffold of the series. The most potent inhibitor was compound **14** with a proline (*K*_i_ = 21 μM) in *para*-position relative to *tert*-butyl group, followed by compound **9**, featuring thiophenethyl-substituent (*K*_i_ = 24 μM) in the *meta*-position, and compound **5**, which contains thiocyclopentane in *para* position relative to the *tert*-butyl group (*K*_i_ = 29 μM). The weakest inhibition potency was observed for compound **6**, which bears a thiobenzen substituent in the *meta* position relative to the *tert*-butyl group. Its potency was approximately five times lower than that of the most active compounds mentioned above. Although the series includes a relatively small number of compounds and a comprehensive structure–activity relationship (SAR) analysis cannot be performed, some general trends can be observed. The obtained data suggest that AChE prefers thio-substitunts in the *para* over the *meta* position relative to the *tert*-butyl group on quinone. Specifically, compounds **5** and **3** exhibit approximately 1.7-fold higher inhibition potency compared to their *meta* position counterparts, compounds **4** and **2**, respectively. Moreover, as these analogs are regioisomers at the C(5) and C(6) positions, this suggests that AChE displays a certain degree of regioselectivity, favoring C(5) over to C(6) regioisomers. The observed regioselective preferences suggest that electronic effects contribute to the inhibition potency, likely due to the synchronized electron-donating and electron-withdrawing properties of the *tert*-butyl and alkylthio groups, respectively. These effects appear to be more harmonized in the C(5) isomer. The electron-withdrawing effect of the sulfur atom arises from its higher electronegativity compared to carbon, while its electron-donating character is attributed to the presence of free electrons in conjunction with a π-aromatic system. The alkylthio group is an electron-withdrawing group due to enhanced electronegativity of sulfur compared to the carbon atom and is an electron-donating group due to the presence of free electrons in conjunction with a π-aromatic system of 1,4-benzoquinone. However, overlapping during electron delocalization is not favorable due to the discordance in orbital size, and therefore, the studied thio-groups should be considered electron-withdrawing groups. Compared to galantamine and donepezil [50], drugs used for the treatment of AD, com-pounds **14**, **9**, and **5** (the most potent AChE inhbitors from the series), were about 54 and 1166 times, respectively, less potent as inhibitors of AChE.

The binding mode of the compounds to AChE was evaluated using graphical plots based on the Hunter–Downs equation, specifically the relationship between *K*_i,app_ and substrate concentration (*s*), which allows discrimination among competitive, non-competitive, and mixed types of inhibition (representative plots for all three modes of inhibition are shown in Appendix A). When *K*_i,app_ proportionally depends on the substrate concentration, the compound binds exclusively to the catalytic anionic site (CAS). In contrast, a non-linear relationship indicates simultaneous binding to both CAS and the peripheral aromatic site (PAS) [35]. For non-competitive binding, the inhibitor binds only to the PAS and *K*_i,app_ remains independent of substrate concentration. Considering the potential to influence the formation of amyloid-AChE plaques, neurotoxic aggregates observed in vitro and in vivo in rat hippocampus [51], compounds that interact with amino acids in PAS (i.e., those exhibiting non-competitive and mixed type of inhibition) are preferred. Among the compounds tested as AChE inhibitors, only compounds **5**, **7**, and **15** bind exclusively to the CAS and thus are unlikely to interfere with the interaction between AChE PAS and amyloid peptides.

The binding modes predicted from the graphical Hunter–Downs evaluation were confirmed utilizing a flexible molecular docking approach to model the interactions of compounds **5**, **9**, and **10** within the AChE active site gorge. According to Hunter–Downs plot, compounds **5**, **9**, and **10** were classified as competitive, non-competitive, and mixed-type inhibitors, respectively. Molecular docking predicted that compound **5** predominantly occupies the lower part of the AChE active site gorge (Figure 2A), corresponding to the catalytic anionic site (CAS). It engages in non-bonding interactions with two residues from the catalytic triad (Ser203 and His447), as well as Trp86 and Tyr337 from the choline-binding site and Phe297 from the acyl pocket. Compound **5** is also predicted to interact with the peripheral anionic site (PAS) residues Tyr124 and Tyr341, but not with the more solvent-exposed PAS residues Tyr72 and Trp286. In contrast, mixed-type inhibitor 10 is predicted to occupy mainly the upper part of the AChE active site gorge (Figure 2B) and interact with PAS residues Tyr124, Tyr341, and Trp286. It also interacts with at least one CAS residue, namely the choline-binding site residue Tyr337, and notably, its *tert*-butyl group protrudes into CAS. Finally, the non-competitive inhibitor 9 is predicted to bind exclusively at the entrance of the AChE gorge (Figure 2C), interacting only with PAS residues Tyr72, Trp286, and Tyr341, while directing its phenethyl ring toward the solvent. Importantly, in all three cases, the compounds are predicted to interact with neighboring residues via their sulfur atoms (Appendix A), highlighting the significance of non-covalent interactions involving sulfur for stabilizing the compounds within the AChE active site [43,52]. Two-dimensional representations of the modeled complexes for AChE with the tested compounds are provided in the Appendix A (Appendix A).

In the case of BChE, the tested compounds inhibited the enzyme with *K*_i_ constant in 5.2–228 μM range. The most potent inhibitors were compounds **3**, containing a 2-hydroxyethylthio substituent, and compound **5**, bearing thiocyclopenthyl group in the *para* position in relation to the *tert*-butyl group. Their respective *K*_i_ values were approximately seven times lower than that of compound **1**. The least potent BChE inhibitor was compound **12**, which has a secondary amine substituent. Interestingly, BChE appears to display regioselectivity similar to AChE, favoring the *para* position (C(5)-regioisomer) over the *meta* position (C(6)-regioisomer). Namely, compounds **3** and **5**, with substituents in the *para* position were about five times more potent than their *meta*-position counterparts, compounds **2** and **4**. To the best of our knowledge, the regioselectivity in the interaction of C(5)-and C(6)-benzoquinone regioisomers with AChE or BChE has not been previously reported in the literature.

To further explain BChE inhibition potency of tested compounds, we again employed molecular docking. The modeled complexes of BChE with the most potent inhibitors, compounds **5** (Figure 3A) and **3** (Figure 3B), indicate that both compounds predominantly occupy the lower part of the BChE active site gorge. This positioning allows them to establish non-covalent interactions with residues from the catalytic triad—Ser198 and His438 in case of inhibitor **3** and Ser198 in case of inhibitor **5**. Similarly, to AChE inhibition, compounds are stabilized within BChE active site through non-covalent interactions involving sulfur atom (Appendix A). In contrast, the significantly less potent BChE inhibitor **10**, although predicted to occupy the lower part of the BChE active site gorge (Figure 3C), lacks both non-covalent interactions with catalytic triad residues and non-covalent interactions involving sulfur atom, which likely contributes to its reduced inhibitory potency.

In general, the tested compounds showed a preference for binding to BChE over AChE, with selectivity indices (SI) ranging from 1.6 to 7.6 in favor of BChE. For compound 12, an SI could not be determined, as it did not inhibit AChE at any of the tested concentrations. The highest selectivity was observed for compound **3** (an alkylthio derivative) and compound **6** (an arylthio derivative). Two-dimensional representations of the modeled complexes between BChE and the tested compounds are provided in Appendix A.

Notably, in comparison to galantamine and donepezil [50]—both standard drugs used in AD therapy—compounds **3** and **5** were approximately five times less potent inhibitors of BChE.

### 3.2. Antioxidative Power

Oxidative stress, resulting from an imbalance between the production of free radicals and their degradation, is recognized as one of the major risk factors for the onset and progression of AD. As benzoquinones are a group of compounds known for their antioxidant properties [53,54], in this study we evaluated the antioxidant potential of the compounds using two assays based on different mechanisms of antioxidant–radical interactions: electron transfer and hydrogen atom transfer.

The first assay used was the FRAP assay, which assesses the reducing ability of compounds via an electron transfer mechanism [55]. The ability to reduce Fe^3+^ to Fe^2+^ was tested for twelve out of fifteen *para*-benzoquinones, selected based on their AChE and BChE inhibition profiles. Only compounds that inhibited at least one of the enzymes with *K*_i_ values less than 200 μM were included. The antioxidant activity was measured at 50 and 100 μM, concentrations chosen based on their determined *K*_i_ values. The obtained FRAP values (Figure 4) showed that tested benzoquinones exhibited poor to moderate antioxidative activity when compared to standard antioxidants Trolox and BHT. No clear trend in activity was observed between the C(5)- and C(6)-regioisomers. Among all of the tested compounds, compound **1** displayed the highest antioxidant potential, though it was still approximately three times less potent than BHT. In a previous study [56], compounds **1**, **6**, and **11** were evaluated for antioxidant activity using the DPPH assay, which is based on a mechanism involving either electron transfer or hydrogen atom transfer from the antioxidant to the DPPH radical, simulating lipid oxidation inhibition [57]. In that study, all three compounds were found to be approximately twice as potent as ascorbic acid, with compound **1** showing the strongest activity and compound **11** the weakest. These results are consistent with our FRAP findings and support the conclusion that the antioxidant mechanism of these benzoquinones is primarily based on electron transfer.

The ORAC assay was used to evaluate the ability of 1,4-benzoquinone derivatives to prevent oxidative damage caused by generation of ROS. In this assay, the quenching of the fluorescent probe (fluorescein) over time by the peroxyl radicals generated by the Free Radical Initiator Solution was monitored. Since the antioxidant capacity of benzoquinones correlates with the fluorescence decay, the peroxyl radical antioxidant activity of the selected compounds was determined by comparing the antioxidant standard curve of Trolox and was expressed as TE (Trolox equivalents)/µM [58].

Based on their anticholinesterase activity, nine 1,4-benzoquinone derivatives were tested at the 50 µM concentration (Table 2). The highest peroxyl radical antioxidant activity was achieved by compound **7**, bearing a phenylthio substituent in the *meta* position relative to the *tert*-butyl group, whose antioxidant power exceeded that of Trolox. Following 7, compound **15** with *S*-phenylalanin as a substituent located in the *meta* position relative to the *tert*-butyl group also displayed antioxidant activity (43.2 TE/µM) comparable to Trolox. Compounds **3** (with 2-hydroxyethylthio substituent in the *para* position relative to *tert*-butyl group), **14** (with *para*-positioned proline), and unsubstituted **1** exhibited lower antioxidant activity 2–3 times compared to Trolox. The remaining tested compounds were over 4.5 times weaker peroxyl radical antioxidants than Trolox.

### 3.3. BACE1 Inhibition

To evaluate their potential impact on the formation of amyloid-β plaques found in the brains of people with AD, benzoquinones were tested as inhibitors of BACE1, the enzyme responsible for the initial β-amyloid peptide formation. From the series of fifteen compounds, seven compounds (**3**, **5**, **7**, **9**, **10**, **14**, and **15**) were selected based on their ability to inhibit the action of AChE and/or BChE. All of the selected compounds inhibited the activity of BACE1 with percentages of inhibition ranging from 5.7 to 27.4% (Figure 5). The highest inhibition, 27.4% and 26.2%, was observed for compounds **9** and **10**, which bear phenethylthio and 2-naphthylthio substituents, respectively. The fact that the lowest inhibition was obtained with compound **3** (5.7%) with 2-hydroxyethylthio as a substituent pointed to the importance of compound volume as a contributor to the inhibitory potency of 1,4-benzoquinones. This finding aligns with previous reports where 1,4-benzoquinones featuring longer side chains inhibited BACE1 activity in the medium nM to low µM range [26,59]

### 3.4. Biometal Chelation

The ability of benzoquinones to reduce the production of ROS and the formation of toxic metal–Aβ plaques, metal-dependent processes which occur in AD, was determined for biometal ions Fe^2+^, Cu^2+^, and Zn^2+^ [60,61]. Absorbance spectra of mixtures containing benzoquinones and biometals were recorded at 1, 60, and 120 min after mixing. For all tested 1,4-benzoquinones, a decrease in absorbance intensity upon the addition of biometals was observed, indicating the formation of a biometal-compound complex (an example experiment is shown in Figure 6A). Spectra recorded 60 min after the incubation period were chosen for differential spectra analysis, as no further changes were detected beyond this time point. The differential spectra (Figure 6B) confirmed the chelating ability of the benzoquinones.

### 3.5. Druggability of Compounds

The druggability of the compounds was assessed in silico using widely accepted computational methods and platforms. However, these studies rely on existing databases that may not include compounds structurally similar to those under investigation. Therefore, the obtained druggability results should be considered preliminary and serve primarily as a computer-based exploration of potential interactions.

#### 3.5.1. Oral Bioavailability

Based on the calculated physicochemical properties crucial for the determination of the oral bioavailability of potential drugs based on extended Lipinski’s rule of five (Mw, log*P*, HBD, HB, RB, and PSA), all compounds, except compound **10**, should be able to cross the blood–brain barrier after oral consumption and thus potentially act as central nervous system active drugs (CNS) (Appendix A). Since compound **10** does not comply with Lipinski’s rule because its lipophilicity is slightly above recommended log*P* < 5, it is expected that 10 would be less active in CNS compared to the rest of the compounds from the series, although existence of drugs “beyond the rule of 5” suggests that compounds with one or two violations of the rule of five could still be bioactive [62].

#### 3.5.2. Human Intestinal Absorption

The percentage of benzoquinone derivatives predicted to be absorbed through the human intestine was estimated in silico using the pkCSM prediction model [46]. According to the model, with the absorption percentages in the range between 88.4 and 96.1 (Appendix A), all of the compounds would likely be very well absorbed in human intestines.

#### 3.5.3. Toxicity Profile of Compounds

The toxicity profile of benzoquinones was estimated using pkCSM predictive models for Ames toxicity, hepatotoxicity, and Minnow toxicity (Table 3). According to the Ames predictive model, all compounds were classified as non-toxic. The hepatotoxicity prediction model rated most of the compounds as non-hepatotoxic; only compound 15 was estimated to be hepatotoxic. According to the Minnow toxicity model, all compounds, except for 9 and 10, which had an estimated LC_50_ = 0.33 and 0.05 mM, respectively, were non-toxic. However, when comparing these LD50 values with the values of *K*_i_ constants determined for AChE and BChE, it was observed that compound **9** had fourteen times lower *K*_i_ value than estimated toxicity, implying it should not affect its possibility of application as a therapeutic, and that only compound **10** can be considered as the compound with very low therapeutically potential. Compounds **1**, **6,** and **11** were previously tested [56] on three human cell lines: MRC-5 (healthy human fetal lung cell line; ECACC No. 84101801), HS 294T (cancer cell line—human melanocytes; ATCC HTB-140), and A549 (human lung cancer cell line; ATCC CCL-185); their results displayed that all except one of the tested compounds from that series showed stronger cytotoxic effect to MRC-5 than compound **1**. In general, the cytotoxic effects were more pronounced after extended incubation (72 h) [56].

## 4. Conclusions

The low-molecular-weight 1,4-benzoquinone derivatives described in this study, featuring sulfur- or nitrogen-containing substituents, have demonstrated a range of valuable biological activities: anticholinesterase and anti-BACE1 activity, antioxidant potency, and biometal chelating ability. These multifunctional properties suggest that the compounds may serve as promising building blocks for the development of multitarget-directed ligands (MTDLs). By combining these derivatives with established pharmacophores, it may be possible to design hybrid compounds capable of simultaneously modulating several key pathological targets relevant to Alzheimer’s disease. However, this study was conducted on a relatively limited set of compounds, and the broader chemical space remains unexplored. Additionally, toxicity assessments were based solely on in silico predictions, and further experimental validation is necessary. Given their favorable biological activity profiles, predictions of good intestinal absorption and oral bioavailability, future work should focus on expanding the chemical diversity of the series. This could include the introduction of additional heterocyclic substituents and halogen atoms. Furthermore, in vitro cellular assays should be employed to establish a more accurate and comprehensive cytotoxicity profile.

## Figures and Tables

**Figure 1 biomolecules-15-01162-f001:**
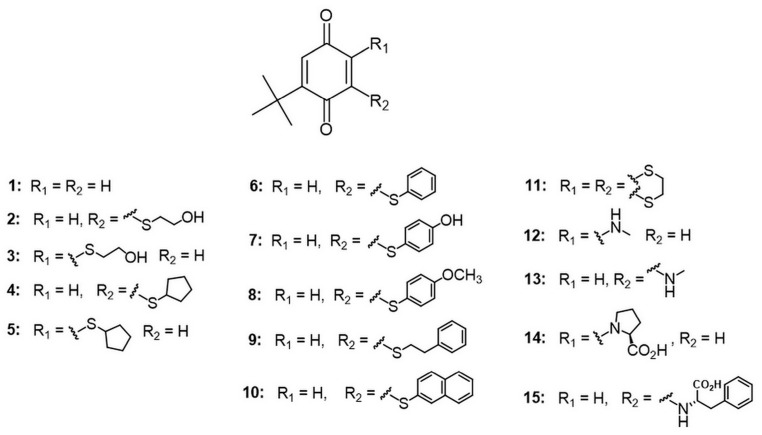
The general structure of the tested compounds.

**Figure 2 biomolecules-15-01162-f002:**
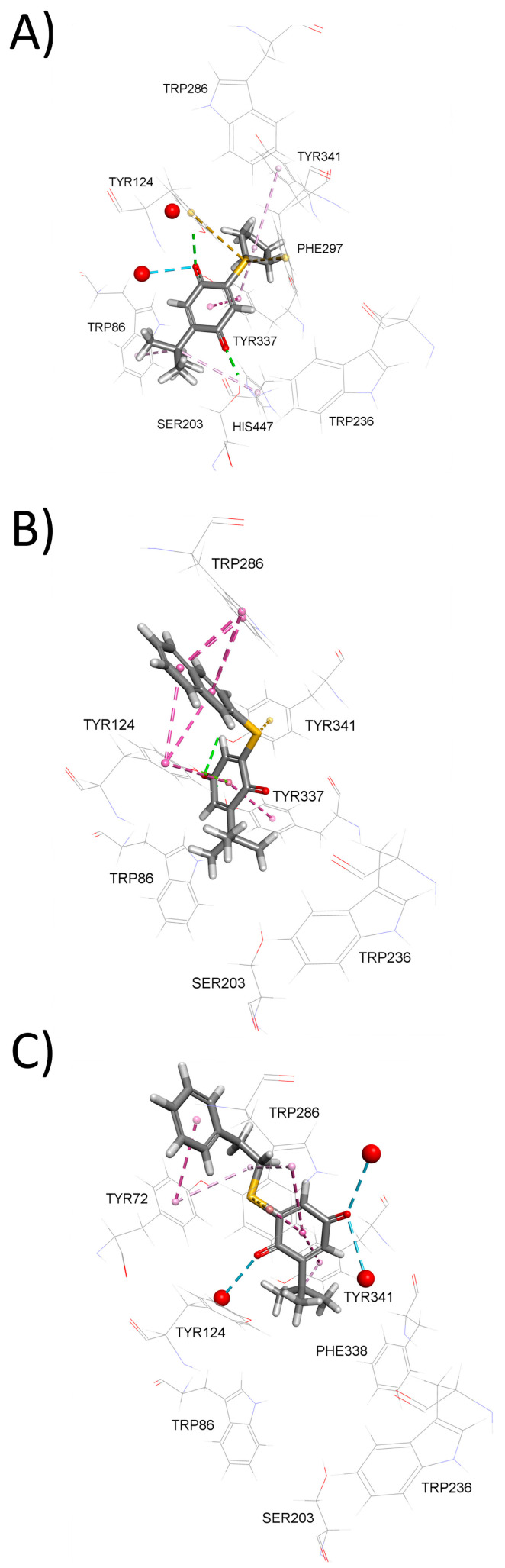
Model complexes of AChE and (**A**) **5**, (**B**) **10** and (**C**) **9**. Dashed lines represent different types of non-bonding intermolecular interactions (magenta—π–alkyl and π–π interactions; orange—electrostatic interaction; green—conventional hydrogen bond; light green—carbon–hydrogen bond; blue—water–hydrogen bond). Red spheres represent conserved water molecules; only water molecules predicted to interact with ligands are shown. For the 2D view of non-bonding interactions, the reader should refer to Appendix A.

**Figure 3 biomolecules-15-01162-f003:**
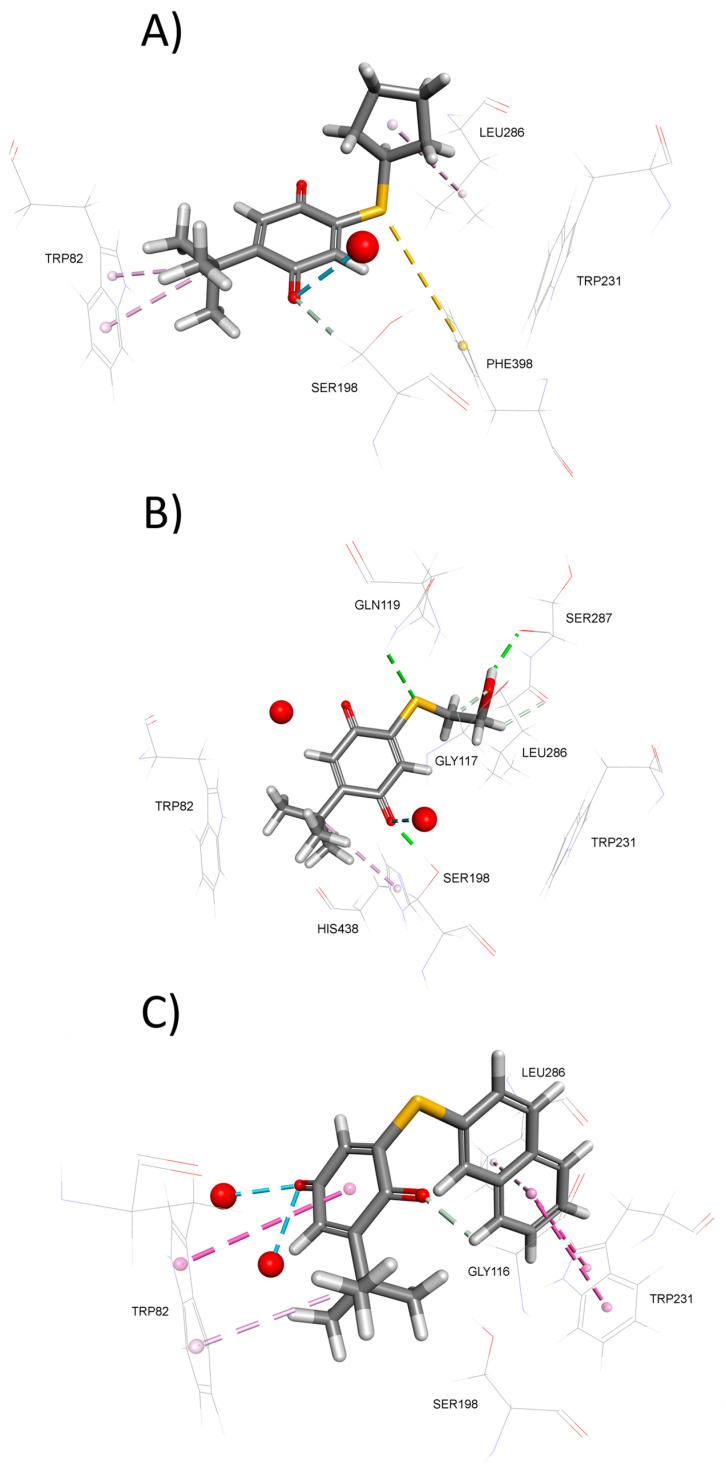
Model complexes of BChE and (**A**) **5**, (**B**) **3**, and (**C**) **10**. Dashed lines represent different types of non-bonding intermolecular interactions (magenta—π–alkyl and π–π interactions; golden—π–sulfur; green—conventional hydrogen bond; light green—carbon–hydrogen bond; blue—water–hydrogen bond). Red spheres represent conserved water molecules; only water molecules predicted to interact with ligands are shown. For the 2D view of non-bonding interactions, the reader is referred to Appendix A.

**Figure 4 biomolecules-15-01162-f004:**
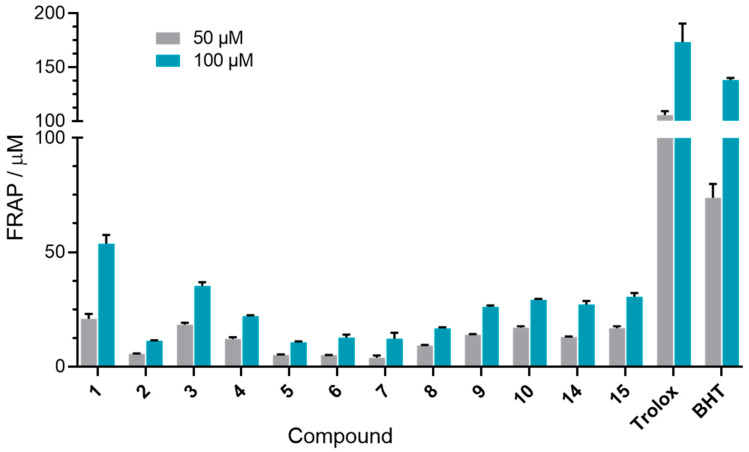
Determined FRAP values. Columns represent the FRAP values for a 50 and 100 μM concentration of each tested 1,4-benzoquinone.

**Figure 5 biomolecules-15-01162-f005:**
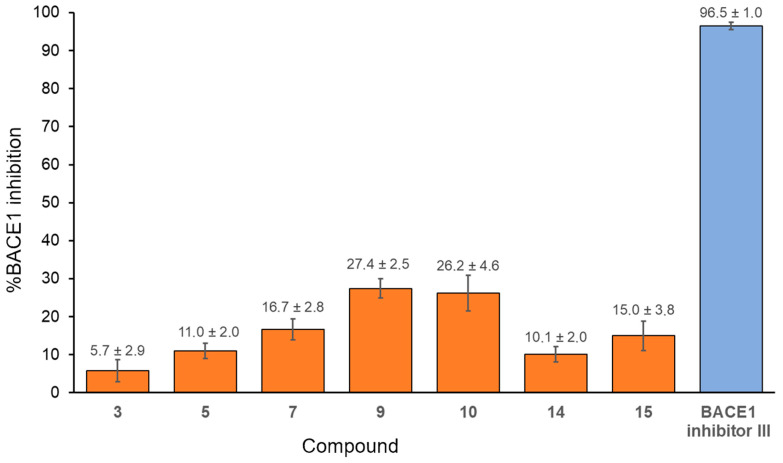
The percentages of BACE 1 inhibition by selected 1,4-benzoquinones. The concentration of benzoquinones was 50 μM, and BACE1 inhibitor III was 5 μM. Presented are values obtained from three independent measurements ± SD.

**Figure 6 biomolecules-15-01162-f006:**
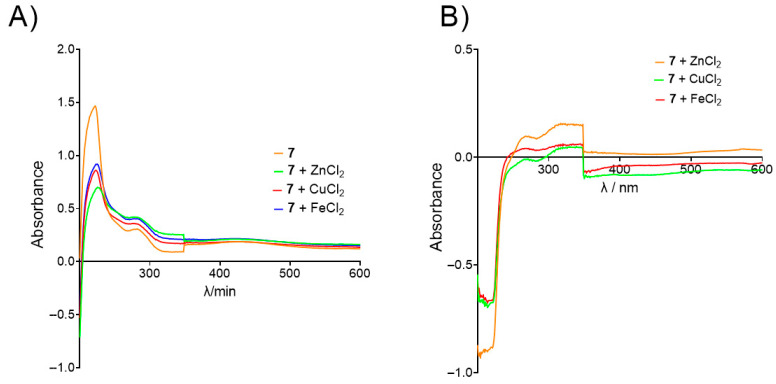
Example of biometal chelation experiment. (**A**) UV/Vis spectra of 7 alone (orange) and in the presence of Zn^2+^ (green), Cu^2+^ (red), and Fe^2+^ (blue) recorded after 60 min of incubation; (**B**) differential spectra of 7-Zn^2+^ (orange), 7-Cu^2+^ (green), and 7-Fe^2+^ complex (red).

**Table 1 biomolecules-15-01162-t001:** Inhibition of cholinesterases. Dissociation constants of enzyme–inhibitor complexes (*K*_i_ ± SE) were determined from at least three experiments.

	AChE	BChE	SI
	*K*_i_/µM	*K*_i_/µM	
**1**	52 ± 6 (n)	55 ± 5	0.95
**2**	80 ± 7 (m)	33 ± 1	2.4
**3**	57 ± 4 (m)	8.8 ± 0.8	6.5
**4**	56 ± 8 (m)	30 ± 6	1.9
**5**	29 ± 5 (c)	5.2 ± 0.7	5.6
**6**	136 ± 13 (n)	18 ± 4	7.6
**7**	37 ± 4 (c)	23 ± 2	1.6
**8**	85 ± 5 (m)	29 ± 3	2.9
**9**	24 ± 1 (n)	25 ± 3	0.96
**10**	30 ± 3 (m)	37 ± 5	0.81
**11**	-	-	-
**12**	n.d.	228 ± 56 *	n.d.
**13**	n.d.	n.d.	-
**14**	21 ± 1 (m)	36 ±2	0.58
**15**	59 ± 5 (c)	34 ± 2	1.7
galantamine [50]	0.52 ± 0.03	1.08 ± 0.08	0.48
donepezil [50]	0.024 ± 0.007	2.33 ± 0.73	0.010

c, n, and m stand for competitive, non-competitive, and mixed-type inhibition, respectively. n.d. stands for non-determined. * The value of *K*_i_ was determined from the 0.05–0.2 mM range of ATCh.

**Table 2 biomolecules-15-01162-t002:** The ORAC values of the tested benzoquinones (50 µM) were determined as TE (Trolox equivalents)µM.

Compound	TE/µM
**1**	16.7 ± 0.1
**3**	23.9 ± 1.6
**5**	11.0 ± 0.3
**7**	≥50
**8**	3.6 ± 0.6
**9**	5.2 ± 0.0
**10**	9.3 ± 0.5
**14**	19.7 ± 1.6
**15**	43.2 ± 1.6

**Table 3 biomolecules-15-01162-t003:** In silico estimated toxicity profile of benzoquinones.

Compound	Model		
	Ames Toxicity	Hepatotoxicity	Minnow Toxicity (LC_50_/mM)
**1**	No	No	30.9
**2**	No	No	20.3
**3**	No	No	34.2
**4**	No	No	1.49
**5**	No	No	2.24
**6**	No	Yes	0.62
**7**	No	No	2.19
**8**	No	No	1.31
**9**	No	No	0.33
**10**	No	No	0.05
**11**	No	No	1.49
**12**	No	No	100
**13**	No	No	100
**14**	No	No	260
**15**	No	Yes	5.20

## Data Availability

The original contributions presented in this study are included in the article/Appendix A. Further inquiries can be directed to the corresponding author.

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
