# Peer review of "Assessment of Biological Activity of Low Molecular Weight 1,4-Benzoquinone Derivatives"

_biomolecules, 2025, doi:10.3390/biom15081162_

Round 1
Reviewer 1 Report
Comments and Suggestions for Authors
The manuscript addresses an interesting topic from a multidisciplinary perspective; however, several aspects require clarification, improved reporting, and revisions to enhance scientific rigor and readability. While reviewing the manuscript, I found that the English language used was often difficult to follow. In particular, the writing lacks clarity and fluidity in several sections, which may hinder reader comprehension. I would recommend a thorough language revision to improve the overall flow and readability.
Below are my detailed comments and suggestions:
Introduction:
- Lines 36 & 38: Please provide appropriate references for these statements.
- Line 41: The introduction currently relies on a single reference from 2018. This is insufficient to support the breadth of information presented. Please include more recent and relevant literature to strengthen the background section.
- Up to line 54, the authors mention only two hypotheses for Alzheimer’s disease (AD). However, these are not the only existing hypotheses. A brief acknowledgment of other major hypotheses in the field is recommended to present a more balanced and comprehensive introduction and to support the concept that AD is a multi-target disease that requires a multi-target treatment approach.
Results and Discussion (and Figures):
- Line 91: Please specify whether the data reported from other groups refers to in vivo or in vitro
- Figure 1: The figure quality is inadequate, particularly regarding the visibility of R groups. Please improve the resolution and clarity of this figure.
- Line 159 (Title of section 2.3): This title does not follow the same formatting style as the other section titles. Please ensure consistency throughout the manuscript.
- Table 1: The reported Ki value for BChE inhibition by compound 12 is not statistically reliable. I suggest either reporting this value as “n.d.” (not determined) or “>350,” or performing additional experiments to better define this parameter.
- Line 298: The range for Ki should be corrected to 20 (21 – 1 for compound 14) up to 149 (136 + 13 for compound 6).
- Lines 309–314: The sentence structure is unclear and appears to be missing punctuation. Please revise for clarity.
- Lines 309–319: If possible, please provide relevant literature to support the general observations reported in this section.
- Lines 321–333: The type of enzyme inhibition (competitive, non-competitive, mixed) is discussed. However, the discussed graphical representations of these results should be provided, if not in the main text, at least as supplementary material.
- Lines 334–336: Please clarify whether docking data for all studied compounds exist. If so, these should be made available in the supplementary material. If docking was performed for only three compounds, please provide a clear justification for this choice.
- Compound Number Reporting: Throughout the manuscript, compound numbers should be consistently reported in bold to avoid confusion.
- Figure 2: The overall quality is low, and the visible frame around each panel is distracting. Please improve the figure resolution and remove unnecessary frames.
- Line 363: The Ki range should be corrected to 4.5 (5.2 – 0.7 for compound 5) up to 523 (372 + 151 for compound 12). If the authors follow the earlier suggestion of reporting compound 12 as “n.d.” or performing additional experiments, the upper range should reflect the highest valid Ki + SD.
- Lines 362–370: Please rewrite this paragraph using clearer English and improved grammar to convey the concepts more effectively.
- Lines 371–383: The discussion lacks comparison to relevant literature. Please integrate your findings with existing studies in the field.
- Figure 3: As with Figure 2, the image quality is low, and panel frames are visible. Please improve the figure accordingly.
- Lines 409–413: The two consecutive sentences are contradictory. First, it is stated that “some compounds show activity comparable to BHT.” In contrast, the second sentence claims that “compound 1 showed the highest antioxidant power, about three times less potent than BHT.” Please rephrase and clarify these statements to avoid confusion.
- Figure 4: The x-axis should clearly indicate that it refers to the tested compounds, not numerical values.
- Lines 417–433: This section only reports observational data. Please provide literature comparisons and hypotheses to explain the findings.
- Line 439 (and throughout the text): When referring to the number of compounds, please spell out numbers in full (e.g., seven instead of 7) to avoid confusion between the quantity of compounds and compound identifiers.
- Figure 5: The quality of this figure is insufficient. Please report standard deviations (SD) for each column and clarify which concentration of the control was used. There is also an inconsistency between the caption and the figure regarding the control's label (inhibitor II vs inhibitor III). Please correct this.
- Lines 454–462: It is unclear why Figure 6 reports only the behavior of compound 7. Please provide a rationale for this choice.
- Figure 6: The quality is very low, and the color scheme makes it difficult to distinguish between different conditions (e.g., brown for compound 7 alone, red or purple for metal complexes). Please improve both resolution and color contrast.
- Line 480: In silico should be italicized.
Suggested Experiment: To increase the scientific value of the study, I recommend complementing the current data with at least one experiment in mammalian cells. While it is not necessary for the model to be an AD model, a neuronal cell line would significantly enhance the toxicity assessment and overall impact of the work.
Conclusions:
The conclusions are too general. Please acknowledge the limitations of the current work and explicitly state that additional studies (e.g., in-cell assays as suggested) are needed to validate the findings.
Comments on the Quality of English Language
While reviewing the manuscript, I found that the English language used was often difficult to follow. In particular, the writing lacks clarity and fluidity in several sections, which may hinder reader comprehension. I would recommend a thorough language revision to improve the overall flow and readability.
Author Response
Authors would like to thank Reviewer 1 for comments and valuable suggestions.
Introduction:
Comment 1: Lines 36 & 38: Please provide appropriate references for these statements.
Respond 1: The reference 1 was added.
Comment 2: Line 41: The introduction currently relies on a single reference from 2018. This is insufficient to support the breadth of information presented. Please include more recent and relevant literature to strengthen the background section.
Response 2: The references 2-4 are added.
Comment 3: Up to line 54, the authors mention only two hypotheses for Alzheimer’s disease (AD). However, these are not the only existing hypotheses. A brief acknowledgment of other major hypotheses in the field is recommended to present a more balanced and comprehensive introduction and to support the concept that AD is a multi-target disease that requires a multi-target treatment approach.
Response 3: Three additional hypothesis are added: “However, AD is a complex, multifactorial condition whose pathogenesis remains only partially understood. In addition to the previously described pathophysiological mechanisms—most notably those contributing to dementia as the hallmark clinical manifestation—a variety of additional pathological processes have been identified, giving rise to new hypotheses regarding the origins and progression of the disease. One such theory is the tau hypothesis, which proposes that abnormal hyperphosphorylation of the tau protein leads to its dissociation from microtubules, resulting in microtubule destabilization, subsequent degradation, and the formation of neurofibrillary tangles (NFTs). The accumulation of NFTs disrupts the intracellular transport of nutrients and essential molecules, ultimately impairing synaptic function and contributing to neuronal dysfunction and cognitive decline [10,11]. According to the oxidative stress hypothesis, the overproduction of reactive oxygen species (ROS) is a major contributing factor to AD pathogenesis, due to the brain’s increased vulnerability to oxidative damage. ROS can impair neuronal function by damaging essential cellular biomolecules such as lipids, proteins, and nucleic acids [12]. Numerous studies strongly suggest that neuroinflammation, characterized by the activation of glial cells, particularly microglia and astrocytes, plays a significant role in progression of AD. Namely, in response to the presence of Aβ and tau aggregates in the brain, microglia become activated, migrate to amyloid plaques, and attempt to clear Aβ via phagocytose. However, the immune response, if prolonged, may become detrimental, contributing to synaptic dysfunction and neuronal damage [13]”
Results and Discussion (and Figures):
Comment 4: Line 91: Please specify whether the data reported from other groups refers to in vivo or in vitro
Response 4: The reports of other groups of researchers that are listed in the cited references refer to in vivo or in vitro experiments; one part of the papers presents the results of in vitro experiments, while some authors presented the results of in vivo experiments. This clarification is inserted in the text as follows: “Studies have demonstrated that incorporating the 1,4-benzoquinone core in various molecular scaffolds can yield compounds with potent biological activity relevant to AD. These include strong inhibitory effects on AChE, BChE, and BACE1; the ability to prevent or reverse Aβ aggregation; biometal chelation and regulation of biometal-free and biometal-bound Aβ aggregation, showing promising efficacy in both in vitro and in vivo studies [23–29].]".
Comment 5: Figure 1: The figure quality is inadequate, particularly regarding the visibility of R groups. Please improve the resolution and clarity of this figure.
Response 5: The quality of the Figure 1 is improved, as well as visibility of R groups.
Comment 6: Line 159 (Title of section 2.3): This title does not follow the same formatting style as the other section titles. Please ensure consistency throughout the manuscript.
Response 6: Title of the section 2.3 is italicized to be consistent with other section titles.
Comment 7: Table 1: The reported Ki value for BChE inhibition by compound 12 is not statistically reliable. I suggest either reporting this value as “n.d.” (not determined) or “>350,” or performing additional experiments to better define this parameter.
Response 7: We recalculate the value of Ki for 0.05 – 0.2 mM range of ATCh concentration, as the main reason for high value of SEM were big differences in Ki,app determined for 0.35 mM ATCh determined for high concentrations of compound 12. The recalculated value of Ki is 228 ± 56 mM.
Comment 8: Line 298: The range for Ki should be corrected to 20 (21 – 1 for compound 14) up to 149 (136 + 13 for compound 6).
Response 8: In Table 1 we presented Ki values ± SE determined from at least three experiment for each of compound. Since we discussed the values of Ki constants in the manuscript, and not their variability in the population or measurements, the range of their values is given, not including their standard errors.
Comment 9: Lines 309–314: The sentence structure is unclear and appears to be missing punctuation. Please revise for clarity.
Response 9: Missing punctuation is added and the sentence is revised ad follows: “Namely, inhibition potency of compounds 5 and 3 is about 1.7 times higher compared to its meta position peers, and compounds 3 and 2.”
Comment 10: Lines 309–319: If possible, please provide relevant literature to support the general observations reported in this section.
Response 10: To best of our knowledge, the regioselectivity of AChE or BChE in the interaction of C(6)-C(7)- regioisomers of benzoquinone has not been described in the literature so far. This statement is added to the ed of the paragraph.
Comment 11: Lines 321–333: The type of enzyme inhibition (competitive, non-competitive, mixed) is discussed. However, the discussed graphical representations of these results should be provided, if not in the main text, at least as supplementary material.
Response 11: A representative graphical presentation for all three modes of inhibition are now presented in Figure S1, and the info about is inserted in the sentence “The binding mode of the compounds to AChE was evaluated using graphical plots based on the Hunter-Dawns equation, specifically the relationship between Ki,app and substrate concentration (s), which allows discrimination among competitive, non-competitive and mixed types of inhibition (representative plots for all three modes of inhibition are shown in Figure S1).
Comment 12: Lines 334–336: Please clarify whether docking data for all studied compounds exist. If so, these should be made available in the supplementary material. If docking was performed for only three compounds, please provide a clear justification for this choice.
Response 12: Docking was performed for all compounds that inhibited AChE and/or BChE. 2D presentation of model complexes for AChE and tested compounds can be found in SI, Figures S2−S5.
Comment 13: Compound Number Reporting: Throughout the manuscript, compound numbers should be consistently reported in bold to avoid confusion.
Response 13: Corrected.
Comment 14: Figure 2: The overall quality is low, and the visible frame around each panel is distracting. Please improve the figure resolution and remove unnecessary frames.
Respond 14: Corrected as suggested.
Comment 15: Line 363: The Ki range should be corrected to 4.5 (5.2 – 0.7 for compound 5) up to 523 (372 + 151 for compound 12). If the authors follow the earlier suggestion of reporting compound 12 as “n.d.” or performing additional experiments, the upper range should reflect the highest valid Ki + SD.
Response 15: The presented Ki values ± SE were determined from at least three experiment for each of compound. Since we discussed the values of Ki constants in the manuscript, and not their variability in the population or measurements, the range of their values is given, not including their standard errors.
Comment 16: Lines 362–370: Please rewrite this paragraph using clearer English and improved grammar to convey the concepts more effectively.
Response 16: The paragraph is rephrased as follows: “In case of BChE, the tested compounds inhibited the enzyme with Ki constant in 5.2–228 mM range. The most potent inhibitors were compounds 3, containing a 2-hydroxyethylthio substituent, and compound 5, bearing thiocyclopenthyl group in para position with respect to tert-butyl group. Their respective Ki values were approximately seven times lower than that of compound 1. The least potent BChE inhibitor was compound 12, which has a secondary amine substituent. Interestingly, BChE appears to display regioselectivity similar to AChE, favoring the para position (C(5)-regioisomer) over meta position (C(6)-regioisomer). Namely, compounds 3 and 5, with substituents in the para position were about five times more potent than their meta-position counterparts, compounds 2 and 4.”
Comment 17: Lines 371–383: The discussion lacks comparison to relevant literature. Please integrate your findings with existing studies in the field.
Response 17: To best of our knowledge, the regioselectivity of AChE or BChE in the interaction of C(6)-C(7)- regioisomers of benzoquinone has not been described in the literature so far. This statement is added at the end of the paragraph.
Comment 17: Figure 3: As with Figure 2, the image quality is low, and panel frames are visible. Please improve the figure accordingly.
Response 17: Corrected as suggested.
Comment 18: Lines 409–413: The two consecutive sentences are contradictory. First, it is stated that “some compounds show activity comparable to BHT.” In contrast, the second sentence claims that “compound 1 showed the highest antioxidant power, about three times less potent than BHT.” Please rephrase and clarify these statements to avoid confusion.
Response 18: The sentence “Although they all have lower antioxidative activity than Trolox and BHT, some show activity comparable with BHT at 50 µM concentration.” is deleted.
Comment 19: Figure 4: The x-axis should clearly indicate that it refers to the tested compounds, not numerical values.
Respond 19: Corrected. The “Compounds” was inserted below the x-axis.
Comment 20: Lines 417–433: This section only reports observational data. Please provide literature comparisons and hypotheses to explain the findings.
Response 20: The paragraph was added as follows: “In a previous study [56], compounds 1, 6, and 11 were evaluated for antioxidant activity using the DPPH assay, which is based on a mechanism involving either electron transfer or hydrogen atom transfer from the antioxidant to the DPPH radical, simulating lipid oxidation inhibition [57]. In that study, all three compounds were found to be approximately twice as potent as ascorbic acid, with compound 1 showing the strongest activity and compound 11 the weakest. These results are consistent with our FRAP findings and sup-port the conclusion that the antioxidant mechanism of these benzoquinones is primarily based on electron transfer.”
Comment 21: Line 439 (and throughout the text): When referring to the number of compounds, please spell out numbers in full (e.g., seven instead of 7) to avoid confusion between the quantity of compounds and compound identifiers.
Response 21.: Corrected as suggested.
Comment 22: Figure 5: The quality of this figure is insufficient. Please report standard deviations (SD) for each column and clarify which concentration of the control was used. There is also an inconsistency between the caption and the figure regarding the control's label (inhibitor II vs inhibitor III). Please correct this.
Response 22: Values of SD are added in the figure, and the inhibitor concentration and label are corrected.
Comment 23: Lines 454–462: It is unclear why Figure 6 reports only the behavior of compound 7. Please provide a rationale for this choice.
Response 23: Figure 6 is an example of metal chelation experiment. This clarification is added in manuscript (“For all tested 1,4-benzoquinones, a decrease in absorbance intensity upon the addition of biometals was observed, indicating the formation of a biometal‒compound complex (an example experiment is shown in Figure 6A).”) and in figure caption (“Example of metal chelation experiment. (A) UV/Vis spectra of 7 alone (brown) and in the presence of Zn2+ (green), Cu2+ (red) and Fe2+ (purple) recorded after 60 minutes of incubation (B) differential spectra of 7-Zn2+ (green), 7-Cu2+ (red) and 7-Fe2+ complex (purple).”).
Comment 24: Figure 6: The quality is very low, and the color scheme makes it difficult to distinguish between different conditions (e.g., brown for compound 7 alone, red or purple for metal complexes). Please improve both resolution and color contrast.
Response 24: The color contrast and resolution are adjusted.
Comment 25: Line 480: In silico should be italicized.
Response 25: Corrected.
Comment 26: Suggested Experiment: To increase the scientific value of the study, I recommend complementing the current data with at least one experiment in mammalian cells. While it is not necessary for the model to be an AD model, a neuronal cell line would significantly enhance the toxicity assessment and overall impact of the work.
Response 26: Compounds 1, 6 and 11 were previously tested (Đorđević et al, J. Serb. Chem. Soc. 2022) on three human cell lines: MRC-5 (healthy human fetal lung cell line; ECACC No. 84101801), HS 294T (cancer cell line – human melanocytes; ATCC HTB-140) and A549 (human lung cancer cell line; ATCC CCL-185), and their results displayed that all tested compounds showed stronger cytotoxic effect to MRC-5 than compound 1 with exception of compound 4, which was four times more toxic compound. The effect is more pronounced after a longer incubation period (72 h). This info is added in the manuscript.
Conclusions:
Comment 27: The conclusions are too general. Please acknowledge the limitations of the current work and explicitly state that additional studies (e.g., in-cell assays as suggested) are needed to validate the findings.
Response 27: According to suggestions, the Conclusion section is rephrased as follows: “The low molecular weight 1,4-benzoquinone derivatives described in this study, featuring sulphur- or nitrogen-containing substituents, have demonstrated a range of valuable biological a-tivities: anticholinesterase and anti-BACE1 activity, antioxidant potency and biometal chelating ability. These multifunctional properties suggest that the compounds may serve as promising building blocks for the development of multitarget-directed ligands (MTDLs). By combining these derivatives with established pharmacophores, it may be possible to design hybrid com-pounds capable of simultaneously modulating several key pathological targets relevant to Alzheimer's disease. However, this study was conducted on a relatively limited set of compounds, and the broader chemical space remains unexplored. Additionally, toxicity assessments were based solely on in silico predictions, and further experimental validation is necessary. Given their favorable biological activity profiles, predictions of good intestinal absorption, and oral bioavailability, future work should focus on expanding the chemical diversity of the series. This could include the introduction of additional heterocyclic substituents and halogen atoms. Furthermore, in vitro cellular assays should be employed to establish a more accurate and comprehensive cytotoxicity profile.”
Comments on the Quality of English Language
While reviewing the manuscript, I found that the English language used was often difficult to follow. In particular, the writing lacks clarity and fluidity in several sections, which may hinder reader comprehension. I would recommend a thorough language revision to improve the overall flow and readability.
Response: The manuscript was proofread by English language expert, both grammar and style. Due to the extensiveness of the corrections, they are not marked.
Reviewer 2 Report
Comments and Suggestions for Authors
The article "Assessment of Biological Activity of Low Molecular Weight 1,4-Benzoquinone Derivatives" presents a comprehensive study on the potential of 1,4-benzoquinone derivatives as multi-target-directed ligands for Alzheimer’s disease (AD) treatment. The research is well-structured, with clear methodology and detailed results. Below are suggested comments to improve the article, focusing on clarity, scientific rigor, and presentation, while maintaining the integrity of the original work.
Consistency in Terminology:
- The term “biometal” is used inconsistently (e.g., “biometal ions” vs. “metal ions”). Standardize to “biometal ions” throughout for clarity, as this is the term used in AD literature to refer to Fe²⁺, Cu²⁺, and Zn²⁺.
- The abbreviation “AD” is used without being defined at its first occurrence. Define it explicitly in the introduction (e.g., “Alzheimer’s disease (AD)”).
- Line 127–128: Specify the source or purity of DMSO used for preparing stock solutions, as this can affect reproducibility.
- Line 265–266: The statement that compound 11 precipitated in the phosphate buffer could be elaborated. Was this due to solubility issues specific to its structure? This could guide future derivative design.
- Line 279–280: The observation about AChE preferring thio-groups in para vs. meta positions is insightful but could be strengthened by discussing potential steric or electronic effects of the tert-butyl group influencing regioselectivity.
Author Response
Authors would like to thank Reviewer 2 for comments and valuable suggestions.
Comment 1: The term “biometal” is used inconsistently (e.g., “biometal ions” vs. “metal ions”). Standardize to “biometal ions” throughout for clarity, as this is the term used in AD literature to refer to Fe²⁺, Cu²⁺, and Zn²⁺.
Response 1: Corrected through the manuscript.
Comment 2: The abbreviation “AD” is used without being defined at its first occurrence. Define it explicitly in the introduction (e.g., “Alzheimer’s disease (AD)”).
Response 2: The abbreviation is defined in the first sentence of the Introduction.
Comment 3: Line 127–128: Specify the source or purity of DMSO used for preparing stock solutions, as this can affect reproducibility.
Response 3: DMSO was purchased from Gram-mol, Zagreb, Croatia. This info is added in section 2.1.
Comment 4: Line 265–266: The statement that compound 11 precipitated in the phosphate buffer could be elaborated. Was this due to solubility issues specific to its structure? This could guide future derivative design.
Response 4: We didn’t have problem with solubility of the compound, and a stock solution was prepared as 100 mM. But, when we added it to phosphate buffer, it precipitates. We reformulate the sentence to clarify what occurred (“Compound 11 was not tested because it precipitated upon addition to the phosphate buffer used in the assay.”)
Comment 5: Line 279–280: The observation about AChE preferring thio-groups in para vs. meta positions is insightful but could be strengthened by discussing potential steric or electronic effects of the tert-butyl group influencing regioselectivity.
Response 5: The sulphur atom, directly connected to the aromatic core, expresses two types of electronic effects that are mutually opposed. One is electronic-withdrawal (EW) due to enhanced electronegativity compared to the carbon atom, and the second is electron-donating (ED) due to the presence of free electrons in conjunction with a π-aromatic system. However, corresponding occupied orbitals on the sulphur atom bearing electron pairs are larger than those on carbon, and possible overlapping during electron delocalisation is not favourable; therefore, thio-groups should be considered as electron-withdrawal groups (EWG). At the same time, the tert-butyl group has an ED effect due to hyperconjugation, and is generally considered as EDG. From that, C(5) regioisomers have two groups that exhibit a harmonised electronic effect, which are at 180°, enabling maximum addition. From this, qualitatively, the molecular dipole moment of the C(5) regioisomers is higher than that of the C(6) regioisomers. From the observed Ki values of regioisomers, both ChEs have a higher preference for C(5) regioisomers, which leads to the conclusion that electronic distribution contributes to the inhibitory activity of the examined compounds. Currently, we cannot discuss this observation quantitatively, in terms of the directions of dipole moments toward amino acids of the enzymes and quantitative relations between isomers, since we need to perform additional quantum-mechanical calculations, which are beyond the scope of this manuscript and will be addressed in the future.
A corresponding comment is added as follows: “The observed regioselective preferences suggest that electronic effects contribute to the inhibition potency, likely due to the synchronized electron-donating and electron-withdrawing properties of the tert-butyl and alkylthio groups, respectively. These effects appear to be more harmonized in the C(5) isomer. The electron-withdrawing effect of the sulfur atom arises from its higher electronegativity compared to carbon, while its electron-donating character is attributed to the presence of free electrons in conjunction with a π-aromatic system. The alkylthio group is an electron withdrawal group due to enhanced electronegativity of sulphur compared to the carbon atom, and is an electron-donating group due to the presence of free electrons in conjunction with a π-aromatic system of 1,4-benzoquinone. However, overlapping during electron delocalization is not favorable due to the discordance in orbital size and therefore, here studied thio-groups should be considered as electron-withdrawing groups.”.
Reviewer 3 Report
Comments and Suggestions for Authors
The submitted article presents a study on the biological activity of seven different 1,4-benzoquinone derivatives. Only part of the study is based on in vitro analyses, while the remaining portion involves in silico study. In silico analyses represent an important tool in the modeling of molecules with anticipated biological activity. Unfortunately, in the presented work, they do not serve as a preliminary step followed by experimental verification, but rather as a separate analysis of an already synthesized series of compounds. The strength of the article is that it introduces new data on newly developed structures. In this respect, the article provides useful insights into how specific structural features may influence biological activity. By comparing certain compounds with very similar structures, it is possible to make some structure–activity relationship observations (for instant 6,7,8; 6 vs 9; 6 vs 10). In my opinion, such comparisons provide some valuable information for other researchers. Unfortunately, the article also contains several significant shortcomings and does not meet the quality criteria for publication in Biomolecules in its current form. Below, I provide a detailed list of comments and suggestions for improvement.
Comment 1; In the introduction, the Authors explained why they are investigating the activity of 1,4-benzoquinone derivatives, but they did not clarify why the derivatives with substituents containing a sulphur or nitrogen atom were selected. This choice should be justified.
Comment 2; In the enzyme (AChE and BChE) inhibition studies, no reference compound (control) was used, making it impossible to compare the obtained results with a substance of known activity. Such data are required if the article is to be considered for publication, as they provide a necessary point of reference for evaluating the effectiveness of the tested compounds. Biomolecules requires that authors publish all experimental controls.
Comment 3; The manuscript lacks detailed information regarding the experimental procedures. For example, products were isolated using preparative thin-layer chromatography. However, preparative TLC conditions are incomplete, with no mobile phase specified, and the conditions for LC-MS analysis are not described. The full experimental details must be provided such that the results can be reproduced. The purity of the compounds used in the study was also not reported.
Comment 4; The manuscript lacks statistical analysis of the data, which should be added to support the findings.
Comment 5: Table 1 – The title of the table is overly long; part of the information could be included in the table footnotes instead.
Comment 6: In the Material and methods section -The methodology does not describe how the selectivity index was calculated. The Table 1 shows SI. There is no information about what SI is in the explanations.
Comment 7: Line 398 - there is a statement: ” benzoquinones are a group of compounds known for their ability to act as antioxidants,” but no reference to any literature sources is provided.
Comment 8: Line 410 - there is a statement: "some show activity comparable with BHT at 50 μM concentration. “Some” - this observation is quite optimistic looking at Figure 1. Please clarify.
Comment 9: The authors refer to concentrations of substances causing BACE1 inhibition, but these concentrations are 10x lower than the inhibitor used. Please explain why the inhibitor (control) and the analyzed substances were tested at different concentrations. Comparison of other concentrations gives misleading information. Please justify this choice, because using concentrations of the analyzed substances such as the control could probably show their lack of activity.
Comment 10: Figure 5. The figure has the caption BACE1 inhibitor III, and the caption under the figure indicates BACE1 inhibitor II. Please enter correctly.
Comment 11: "Druggability of compounds" – the Authors state that they analyze "druggability", but they do not refer to whether the data obtained from in silico experiments indicate sufficient activity of the compounds to be considered as potential medicinal substances -drugs. It should be remembered and emphasized more strongly that these are in silico studies and it is necessary to thoroughly verify their probability. The authors are examining, among others, quite simple analyses such as the AMES test. When examining compounds with a new structure, it is not always possible to rely on the results obtained for compounds from databases. A lot depends on the database. In silico studies are a good starting point for research, allowing for preliminary analyses and computer-based exploration of potential interactions. Performing such an analysis on compounds in vitro would allow for verification of in silico studies. Additionally, when examining compounds, e.g. in the AMES test, whether their hepatoxicity can be referred to a specific concentration of the compound or a range of concentrations at a specific concentration and compared to recognized mutagenic or cytotoxic substances. If the authors decided to use in silico tests, this should be emphasized more strongly and the limitations of the studies should be presented.
Comment 12: In Supplementary materials, in several places, including in part 1.2. Synthetic procedure, a marking appears indicating the lack of connection between the content and the literature reference “Error! Bookmark not defined." Missing literature references should be entered.
Comment 13: In Figure S3. In the supplement – there are different names of compounds than in the article.
Author Response
Authors would like to thank Reviewer 3 for comments and valuable suggestions.
Comment 1; In the introduction, the Authors explained why they are investigating the activity of 1,4-benzoquinone derivatives, but they did not clarify why the derivatives with substituents containing a sulphur or nitrogen atom were selected. This choice should be justified.
Response 1: We thank the reviewer for this comment. The compounds in this research belong to the broader groups of derivatives of 1,4-benzoquinones, which contain sulphur or nitrogen substituents, and show moderate to promising inhibitory activity against human cancer cell lines. According to data from the literature, 1,4-benzoquinones and other quinone congeners exhibit promising inhibitory activity against ChEs. However, most of them have complex substituents, or they are symmetrical bisubstituted derivatives. With selected derivatives, we aim to investigate whether simpler 1,4-benzoquinone derivatives featuring less complex substituents also exhibit activity and evaluate their potential as CNS agents, as we mentioned in the Introduction. Additionally, substituents bonded to the 1,4-benzoquinone system by sulphur and nitrogen substituents have a strong influence on the electron density of the quinone system, depending on the structure of the side chain; therefore, they could influence the corresponding interaction with amino acid groups of the enzyme chain. And finally, substituents with an electron-donating effect make the 1,4-benzoquinone system rich in electrons and hence less susceptible to nucleophilic attack by various nucleophiles in biological systems, making the corresponding derivatives less toxic. On the other hand, corresponding congeners with oxygen have two significant shortcomings: a) they are challenging to synthesise according to current synthetic procedures, and complex reaction mixtures are obtained, making them complicated for purification; b) oxygen substituents due to their electron withdrawing effect makes corresponding 1,4-benzoquinone system more susceptible for nucleophiles in the biological system which is leading cause of toxicity. We selected a set of derivatives that feature characteristic substituents to investigate potential patterns in the influence of substituents on inhibitor activity. We believe that the obtained results confirmed our hypothesis and justify further development of this class of derivatives.
We add additional explanation for the selection of the described compounds, for this study and add supporting references, as follows: “We selected a series of fifteen 1,4-benzoquinone derivatives, each bearing substituents containing sulphur or nitrogen atoms. These heteroatoms were chosen as we hypothe-sized that they could significantly influence the electron density of the quinone core, thereby affecting its interaction with amino acid residues within enzyme active sites. Furthermore, electron-donating substituents are expected to enrich the quinone ring’s electron density, rendering the system less susceptible to nucleophilic attack by biological nucleophiles—an effect that may contribute to reduced toxicity [30–31]. “
Comment 2; In the enzyme (AChE and BChE) inhibition studies, no reference compound (control) was used, making it impossible to compare the obtained results with a substance of known activity. Such data are required if the article is to be considered for publication, as they provide a necessary point of reference for evaluating the effectiveness of the tested compounds. Biomolecules requires that authors publish all experimental controls.
Response 2: Values of Ki constants for AChE and BChE for inhibition by galantamine and donepezile, drugs in use for treatment of Alzheimer’s disease were added, accompanied by corresponding comparison with the most potent inhibitors from the series for AChE and BChE.
Comment 3; The manuscript lacks detailed information regarding the experimental procedures. For example, products were isolated using preparative thin-layer chromatography. However, preparative TLC conditions are incomplete, with no mobile phase specified, and the conditions for LC-MS analysis are not described. The full experimental details must be provided such that the results can be reproduced. The purity of the compounds used in the study was also not reported.
Response 3: Authors thank the reviewer for pointing out this oversight. The missing information about experimental conditions were added in the main text and the SI document as follows: “All reagents and solvents used for the synthesis were purchased from commercial sources (Fluka, Sigma, Aldrich, Merck or Acros Organics). All solvents were distilled be-fore use. Reaction progress was monitored by thin-layer chromatography (TLC) using Supelco TLC aluminium sheets precoated with Silica gel 60, and UV indicator (254 nm). Preparative TLC was performed on Supelco silica gel 60 GF254 with a UV-active indicator and appropriate mobile phase, as described in the corresponding synthetic procedures (details available in the Supplementary Information). NMR spectra were recorded in deu-terochloroform (CDCl3) on a Bruker Avance III (500 MHz instrument for 1H NMR and 125 MHz for 13C NMR). Chemical shifts are reported in parts per million (ppm) using tetra-methylsilane as the internal standard, the coupling constants (J) are given in Hz, and the multiplets are designated as singlet (s), broad singlet (bs), doublet (d), double doublet (dd), triplet (t), multiplet (m). The ESI-MS spectra of the synthesized compounds were recorded on an Agilent Technologies 1200 Series instrument equipped with a Zorbax Eclipse Plus C18 (100 × 2.1 mm i.d., 1.8 μm) column and a DAD detector (190-450 nm) in combination with an Agilent Technologies 6210 Time-Of-Flight LC-MS instrument in positive ion mode with CH3CN/H2O 1/1 with 0.2 % HCOOH as the carrying solvent solution. Samples were dissolved in MeOH (HPLC grade purity). The capillary voltage = 4 kV, gas temperature = 350 °C, drying gas flow rate = 12 L min−1, nebulizer pressure = 45 psi and fragmentor voltage = 70 V were used.
The purity of the examined compounds could not be determined using HPLC methods due to the instability of the compounds under the applied conditions. Although compounds are 1,4-benzoquinones, they are also conjugated carbonyl compounds that easily succumb to nucleophilic attack, especially in the presence of acid, which is typically present under reverse-phase conditions. The consequence is the appearance of multiple signals or a broad signal in the chromatogram, giving the impression that the sample is not pure but is, instead, a mixture of compounds. On the other hand, chromatography under normal-phase conditions (typically silica gel) yields a very broad signal for the compound, which is also not acceptable as evidence of purity. For these reasons, the purities of the examined compounds are based on NMR spectra and HRMS data, along with accompanying chromatograms obtained during analysis. For all compounds, the purity is greater than 95%. We add an appropriate statement in the main document (section 2.1 Materials) and in the SI document (1.1. General information) as follows: “Since the studied 1,4-benzoquinones are also conjugated carbonyl compounds that readily undergo nucleophilic attack, especially in the presence of acid which is usually present in reversed phase conditions, purities of the examined compounds are based on NMR spectra and HRMS data, along with accompanying chromatograms obtained during analysis. For all compounds, the purity is greater than 95%.”
Comment 4; The manuscript lacks statistical analysis of the data, which should be added to support the findings.
Response 4: All determined Ki constants and percentages of inhibition were presented with corresponding SEM or SD values determined from at least three independent experiment, and comparison with drugs in use for AD is presented. As sources of enzymes used in our experiment were recombinant human AChE, purified BChe and commercially available BACE1 no statistical analysis was needed.
Comment 5: Table 1 – The title of the table is overly long; part of the information could be included in the table footnotes instead.
Response 5: The title of the Table was shorted to “Inhibition of cholinesterases. Dissociation constants of enzyme–inhibitor complexes (Ki ± SE) were determined from at least three experiments.”, and the “ c, n and m stand for competitive, non-competitive and mixed type of inhibition, respectively. n.d. stands for non-determined.” Is moved to the table footnotes.
Comment 6: In the Material and methods section -The methodology does not describe how the selectivity index was calculated. The Table 1 shows SI. There is no information about what SI is in the explanations.
Response 6: The selectivity index (SI) was determined from the ratio: SI = Ki(AChE)/ Ki(BChE). This is added to Materials and Methods section.
Comment 7: Line 398 - there is a statement: ” benzoquinones are a group of compounds known for their ability to act as antioxidants,” but no reference to any literature sources is provided.
Response 7: We omitted to provide references. We correct our omission and provide the corresponding references, which are embedded in the main text:
- a) Polyakov, T. Leshina, L. Fedenok, I. Slepneva, I. Kirilyuk, J. Furso, M. Olchawa, T. Sarna, M. Elas, I. Bilkis, L. Weiner, Redox-Active Quinone Chelators: Properties, Mechanisms of Action, Cell Delivery, and Cell Toxicity, Antioxid Redox Signal, 2018, 28, 1394-1403, doi: 10.1089/ars.2017.7406.
- b) X. Ji, X. Liu, M. Li, S. Shao, J. Chang, J. Du, X. Ma, X. Feng, L. Zhu, X. Yu, W. Hu, Study of the Redox Potentials of Benzoquinone and Its Derivatives by Combining Electrochemistry and Computational Chemistry, J. Chem. Educ., 2021, 98, 3019−3025, https://doi.org/10.1021/acs.jchemed.1c00136.
Comment 8: Line 410 - there is a statement: "some show activity comparable with BHT at 50 μM concentration. “Some” - this observation is quite optimistic looking at Figure 1. Please clarify.
Response 8. This observation is rephrased in “The obtained FRAP values (Figure 4) showed that tested benzoquinones exibited poor to moderate antioxidative activity when compared to standard oxidatsTrolox and BHT. No clear trend in activity was observed between the C(5)- and C(6)-regioisomers. Among all of the tested compounds, compound 1 displayed the highest antioxidant potential, though it was still approximately three times less potent than BHT.
Comment 9: The authors refer to concentrations of substances causing BACE1 inhibition, but these concentrations are 10x lower than the inhibitor used. Please explain why the inhibitor (control) and the analyzed substances were tested at different concentrations. Comparison of other concentrations gives misleading information. Please justify this choice, because using concentrations of the analyzed substances such as the control could probably show their lack of activity.
Response 9: The concentration of BACE1 inhibitor III used in experiments was 5 mM, which was the concentration recommended from the BACE1 and BACE1inhibitorIII manufacturer for validation of BACE1 assay. The concentration of compounds was 50 mM, and it was chosen based on values of their Ki constants determined for both cholinesterases.
Comment 10: Figure 5. The figure has the caption BACE1 inhibitor III, and the caption under the figure indicates BACE1 inhibitor II. Please enter correctly.
Response 10: Corrected.
Comment 11: "Druggability of compounds" – the Authors state that they analyze "druggability", but they do not refer to whether the data obtained from in silico experiments indicate sufficient activity of the compounds to be considered as potential medicinal substances -drugs. It should be remembered and emphasized more strongly that these are in silico studies and it is necessary to thoroughly verify their probability. The authors are examining, among others, quite simple analyses such as the AMES test. When examining compounds with a new structure, it is not always possible to rely on the results obtained for compounds from databases. A lot depends on the database. In silico studies are a good starting point for research, allowing for preliminary analyses and computer-based exploration of potential interactions. Performing such an analysis on compounds in vitro would allow for verification of in silico studies. Additionally, when examining compounds, e.g. in the AMES test, whether their hepatoxicity can be referred to a specific concentration of the compound or a range of concentrations at a specific concentration and compared to recognized mutagenic or cytotoxic substances. If the authors decided to use in silico tests, this should be emphasized more strongly and the limitations of the studies should be presented.
Response 11: The Authors agree with the Reviewer, and to added corresponding statement (“The druggability of the compounds was assessed in silico using widely accepted computational methods and platforms. However, these studies rely on existing databases that may not include compounds structurally similar to those under investigation. Therefore, the obtained druggability results should be considered preliminary and serve primarily as a computer-based exploration of potential interactions.”) as a first paragraph of 3.5 section, and also pointed out it in Conclusion section as follows “However, this study was conducted on a relatively limited set of compounds, and the broader chemical space remains unexplored. Additionally, toxicity assessments were based solely on in silico predictions, and further experimental validation is necessary. Given their favorable bio-logical activity profiles, predictions of good intestinal absorption, and oral bioavailability, future work should focus on expanding the chemical diversity of the series. This could include the introduction of additional heterocyclic substituents and halogen atoms. Furthermore, in vitro cellular assays should be employed to establish a more accurate and comprehensive cytotoxicity profile.”
Comment 12: In Supplementary materials, in several places, including in part 1.2. Synthetic procedure, a marking appears indicating the lack of connection between the content and the literature reference “Error! Bookmark not defined." Missing literature references should be entered.
Response 12: Corrected.
Comment 13: In Figure S3. In the supplement – there are different names of compounds than in the article.
Response 13: Corrected.
Round 2
Reviewer 1 Report
Comments and Suggestions for Authors
The authors made the suggested improvements.
Author Response
Comment: The authors made the suggested improvements.
Respond: Thank you for your time and decision.
Reviewer 3 Report
Comments and Suggestions for Authors
The manuscript has been improved. However, in my opinion, before publication in Biomolecules, the basic statistical analysis to compare activity (in vitro) between compounds tested should be add.
Author Response
Reviewer 3 Comment: The manuscript has been improved. However, in my opinion, before publication in Biomolecules, the basic statistical analysis to compare activity (in vitro) between compounds tested should be add.
Response: We appreciate the reviewer's suggestion and fully agree that statistical analysis is an important component in assessing experimental reproducibility and significance. However, we would like to clarify that, due to the experimental design and nature of the data collected, standard statistical comparisons (such as ANOVA or t-tests) would not be meaningful for a couple of reasons. The first limitation is the small sample size per group, which not only limits statistical power but also renders multiple comparison corrections (e.g., Bonferroni) statistically unreliable. Furthermore, in the case of cholinesterase inhibition, not all compounds were tested in identical concentration ranges due to differences in their inhibition activity or solubility of the compounds (compound 13 was inactive at 400 µM; compound 11 precipitated in the used assay conditions). Although such variation does not affect the quality and reliability of the results and conclusions obtained, it introduces additional variability that precludes a direct statistical comparison across the entire compound set.
In assays of BACE1 inhibition and antioxidant assays (FRAP, ORAC), compounds were evaluated at one or two concentrations. These data could be compared, but do not allow robust statistical analysis as inter-compound comparisons lack dose–response relationships.
The main objective of our work was to observe qualitative SAR trends and explore structure-based inhibitory mechanisms (e.g., binding modes, regioselectivity), rather than to conduct a quantitative comparison of efficacy across the entire compound series.
The methodology applied aligns with widely accepted practices in medicinal chemistry, where Ki or IC₅₀ values, determined from kinetic plots or enzyme inhibition curves, are typically reported as individual estimates, not subjected to inter-compound significance testing.
In our paper, we have ensured rigorous data interpretation through careful kinetic experiments, structure-based docking, and comparison to reference standards (galantamine, donepezil, Trolox). These approaches offer strong support for the biological relevance and comparative potency of the compounds studied.